# Banyan: Improved Representation Learning with Explicit Structure

## Abstract

We present Banyan, a model that efficiently learns semantic representations by leveraging an inductive bias towards explicit hierarchical structure. Although typical transformer-based models excel at scale, they struggle in low-resource settings. Recent work on models exploiting explicit structure has shown promise as efficient learners in resource-constrained environments. However, these models have yet to demonstrate truly competitive performance. Banyan bridges this gap, significantly improving upon prior structured models and providing, for the first time, a viable alternative to transformer embeddings for under-represented languages. We achieve these improvements through two key innovations 1) A novel entangled tree structure that resolves multiple constituent structures into a single shared one, explicitly incorporating global context. 2) Diagonalized message passing functions that increase the influence of the inductive bias. Our final model has just 14 non-embedding parameters yet is competitive with baselines many orders of magnitude larger. Banyan outperforms its structured predecessors and competes with large unstructured models across various semantic tasks in multiple languages. Notably, it excels in low-resource settings, highlighting its potential for efficient and interpretable NLP in resource-constrained environments. These results underscore the value of appropriate inductive biases in capturing semantic relationships and open new avenues for efficient, interpretable NLP models.

## 1 Introduction

Semantic representations of text are important for many NLP applications such as retrieval augmented generation (Lewis et al., 2020), question answering, and summarisation (Abdalla et al., 2023; Wang et al., 2022). They are also useful for clustering and organising textual data when labelled training sets are not available. At the time of writing such representations are primarily generated by large scale transformer models (Vaswani et al., 2017). These models are incredibly effective, but training them usually requires scale, both in terms of data and compute.

An alternative approach is to take inspiration from linguistics/cognitive science and explicitly incorporate structured compositions. Put simply, composition states that all you need to understand the semantics of a whole are the meanings of its parts and the structure that dictates how they fit together (Chomsky, 1956; Crain & Nakayama, 1987; Pallier et al., 2011; de Marneffe et al., 2006). This is a very efficient principle, because novel utterances can broken down into familiar parts using systematic rules, rather than having to store the meaning of each utterance individually. It is thought that this principle lets humans generalise from (comparatively) little data and makes us efficient learners (Fodor & Pylyshyn, 1988; Lake et al., 2016; Ito et al., 2022; Wiedemer et al., 2023). In order to explicitly incorporate an inductive bias of this kind we need to change the modelling process somewhat. Rather than keeping all the information flow internal, models must now learn representations for the atomics, operate on (and/or learn) a discrete graph that dictates the mode of combination, and learn functions that control information flow through such a graph. Models of this kind have demonstrated improved language modelling perplexity at cognitively plausible scales (Hu et al., 2021; 2022); better systematic generalisation (Sartran et al., 2022; Murty et al., 2023); and, importantly for this paper, the ability to efficiently acquire semantics (Opper et al., 2023).

Opper et al. (2023) introduce a model called the `Self-StrAE`, which learns to representations which have to explicitly model compositional semantics. This demonstrated very promising performance

while requiring minimal resources. Both in terms of data and model size. Opening the door to investigate whether more compute efficient solutions can be found for learning semantic representations. This would be particularly useful for low resourced languages where relying on scale is not a generally feasible solution. However, the `Self-StrAE`, while promising, still lags behind large scale pre-trained transformers, even in langauges which fall outside of standard pre-training corpora. In this paper we introduce a model called `Banyan`, which significantly improves performance over that of (Opper et al., 2023), while simultaneously achieving even greater resource efficiency. We achieve this by changing the form of the structure optimised to a graph that models global relations between nodes which we call an entangled trees, as well as a message passing regime based on diagonal functions which reduces parameters while producing more expressive representations. Our model, `Banyan`, achieves competitive performance with transformer based baselines, and for the first time represents a low cost yet viable alternative for producing representations for low resource languages, measured using semantic textual similarity (STS) tasks. By leveraging cognitively inspired inductive biases we can achieve performance comparable or better than large scale pre-trained LLMs but with only 14 non-embedding parameters.

## 2 BACKGROUND AND RELATED WORK

Banyan is a graph neural network, specifically a recursive neural network (RvNN), that learns both structure and representations. Before detailing the model, we unpack these terms in this section.

**Recursive Neural Networks:** Like their recurrent cousins, recursive neural networks operate by repeatedly applying a function to update a the network state in an ordered fashion. However, rather than utilising temporal ordering (i.e. over a sequence), RvNNs operate according to some hierarchical structure, typically this given as input and most often it is a binary tree. They can be applied bottom-up (traversing from leaves to root) or top-down (from root to leaves) or both. First popularised by Socher et al. (2011; 2013), they have inspired numerous successor frameworks which differ in how the recursive function is defined. These successors include the Tree-LSTM (Tai et al., 2015), IORNN (Le & Zuidema, 2014; Ji & Eisenstein, 2015) and also Banyan.

**Learning Structure:** RvNNs typically require structure as input, sometimes such structure is available or can be obtained using existing tools, but generally this is quite a limiting factor, because it limits model flexibility. A solution is to incorporate a mechanism within the model that is able to induce the structure during the recursive computation. Prior approaches include the use of differentiable chart parsing (Drozdov et al., 2019; 2020; Hu et al., 2021; 2022), beam search (Ray Chowdhury & Caragea, 2023), continuous relaxation (Chowdhury & Caragea, 2021; Soulos et al., 2024), or reinforcement learning (Havrylov et al., 2019). While successful these solutions can suffer from memory issues and hyperparameter sensitivity. In this paper we adopt the approach of Opper et al. (2023) and utilise representation similarity to dictate merge order. This is both computationally inexpensive and surprisingly effective.

**Semantic Representations of Text:** Systems like Word2Vec (Mikolov et al., 2013) and GLoVe (Pennington et al., 2014) model the semantics of words using the distributional hypothesis (Harris, 1954). This hypothesis states that the context a word is used in defines its meaning. Consequently, representations are learned by setting a context window of some fixed size and then using that to predict the missing word. This approach proved very effective for a long time, but words don't all have one meaning - it changes in context. A natural solution is to use transformers. Initially smaller (relative to today) encoder only models produced poor representations (Reimers & Gurevych, 2019). However, at time of writing transformers with optional contrastive finetuning (Gao et al., 2021) have become the model of choice for producing semantic representations.

**Semantic Representation Learning through Structure:** Transformer embeddings are more successful than static word embeddings because they are allow for flexible contextualisation. The meaning of a word can change in context and will more strongly influenced by certain neighbours rather than others. A transformer can model this phenomenon by routing information between specific tokens via its attention. An alternative approach is to use structure, where rather than attention dictating routing, it is done by an explicit graph. Early work in this area focused on lexical semantics, using dependency parses to determine a more focused context window (Levy & Goldberg, 2014; Vashishth et al., 2019). More recent work by Opper et al. (2023) use constituency parses in order to learn embeddings at both the word and the sentence level. They introduce two variants of their model. StrAE: which takes

structure as input, and Self-StrAE: which learns its own structure with the representations. This later model, the Self-StrAE, is the starting point from which we build Banyan, and will be outlined in more detail in the subsequent section.

## 3 PRELIMINARY: SELF-STRAE

Self-StrAE involves three main components that acts over a sentence $\mathbf{w} = \langle w_n \rangle_{n=1}^{N}$ represented as a sequence of tokens. These are: (a) a procedure to determine which tokens to merge and in what order, (b) message passing functions—composition and decomposition—that merge and split embeddings respectively, and (c) a reconstructive objective that leverages both the induced structure and embeddings. While we refer the reader to Opper et al. (2023) for a detailed description of these components, we briefly recap its operation to provide sufficient background for the development of Banyan (§ 4).

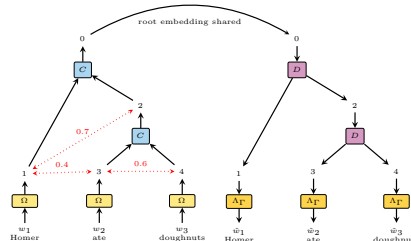

Figure 1: Self-StrAE operation. Red lines indicate cosine similarity. Shared colours imply shared parameters.

At a high-level, Self-StrAE learns representations that both define their own structure and are in turn defined by it. This is achieved by first *embedding* tokens to form an initial frontier using an embedding matrix $\Omega_\Psi$. Next it then takes the adjacent tokens with the highest cosine similarity to each other (*ate* and *doughnuts* in Figure 1) and merges them into a single embedding using a parametric *composition function* $C_\Phi$. This procedure is repeated until the sequence reduces to a single root embedding. The resulting merge history is then treated as the induced binary tree for the sentence. Self-StrAE then traverses back down the structure, recursively splitting embeddings at every node using a parametric *decomposition function* $D_\Theta$ to recover embeddings for the leaves. Finally, the model can optionally use a *dembedding function* $\Lambda_\Gamma$ to predict tokens $\widehat{w}_n$ from these leaf embeddings. Figure 1 illustrates the autoencoding process.

Intuitively, this means that the model starts from random embeddings, and therefore an essentially random merge order. Throughout training, tokens which are often part of the same merges will have their representations drawn together, so the representation reflects what they are likely to compose with. The model can then leverage any regularities to better perform reconstruction. This leads the representations to further reflect likely compositions and consequently increases the regularity in the structure. Ultimately, this leads to representations which must, by virtue of the training procedure, reflect the compositional semantics learned by the model.

For a more formal description of the operation of the model we begin by noting that the model generates *two* sets of embeddings. One set going up from leaves to root, and another coming back down from root to leaves. We denote these $\bar{e}$ and $\underline{e}$ respectively We also note that an embedding is typically viewed as $e \in \mathbb{R}^{U \times K}$ with $K$ independent channels—of particular relevance to the composition and decomposition functions which act independently over the channels. Tokens are denoted as the vertices $w_i \in \Delta^V$ in a $V$-simplex for vocabulary size $V$. All together, the functioning of the model is then characterised by:

$$\Omega_\Psi(w_i) = w_i\,\Psi, \quad \Psi \in \mathbb{R}^{V \times (U \star K)} \tag{1}$$

$$C_\Phi(\bar{e}_i, \bar{e}_{i+1}) = \text{HCAT}(\bar{e}_i, \bar{e}_{i+1})\,\Phi + \phi \quad \Phi \in \mathbb{R}^{2U \times U}, \phi \in \mathbb{R}^{U} \tag{2}$$

$$D_\Theta(\underline{e}_i) = \text{HSPLIT}(\underline{e}_i\,\Theta + \theta) \quad \Theta \in \mathbb{R}^{U \times 2U}, \theta \in \mathbb{R}^{2U} \tag{3}$$

$$\Lambda_\Gamma(\underline{e}_i) = \underline{e}_i\,\Gamma \quad \Gamma \in \mathbb{R}^{(U \star K) \times V} \tag{4}$$

Given the nature of the model, a straightforward objective would be to simply reconstruct the tokens, formulated for sentence $\mathbf{w}$ and prediction $\widehat{\mathbf{w}}$ as $\mathcal{L}_{\text{CE}}(\mathbf{w}, \widehat{\mathbf{w}}) = -\frac{1}{N} \sum_{n=1}^{N} w_n \cdot \log \widehat{w}_n$. An alternate approach developed by Opper et al. (2023) leverages the multi-level structure of the model to define a contrastive objective over a batch of sentences $\{\mathbf{w}_b\}_{b=1}^{B}$ with a total of $M$ nodes (internal + leaves). Noting that the up and down trees share the same underlying structure (modulo reversed edges), this objective draws together corresponding up and down embeddings at a given tree position, whilst pushing away other embeddings across the batch, using the cosine similarity

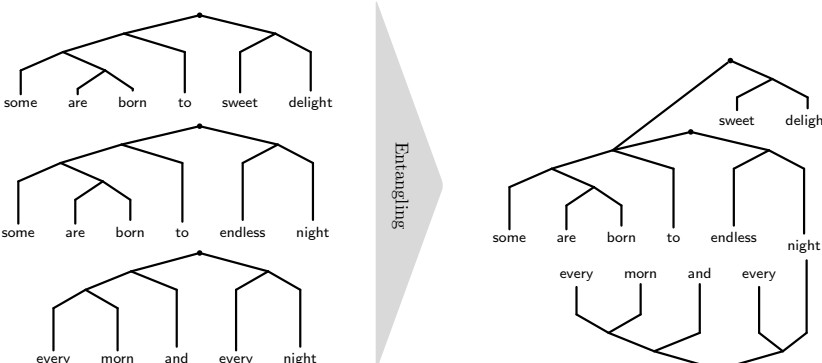

Figure 2: Entangled trees: Example of disjoint trees being transformed into an entangled tree. We leave out denoting functions from Eqs. (1) to (4) to avoid clutter and assume them implicitly present.

metric. Denoting the pairwise similarity matrix $A \in \mathbb{R}^{M \times M}$ between upward embeddings $\langle \bar{e}_i \rangle_{i=1}^M$ and downward embeddings $\langle \underline{e}_i \rangle_{i=1}^M$, and $A_{i\bullet}, A_{\bullet j}, A_{ij}$ the $i^{\text{th}}$ row, $j^{\text{th}}$ column, and $(i,j)^{\text{th}}$ entry of the matrix respectively, the objective is defined as: $\mathcal{L}_{\text{CO}}(\bar{e}, \underline{e}) = \frac{-1}{2M} \sum_{i=1}^M \log\left(\sigma_\tau(A_{i\bullet}) \, \sigma_\tau(A_{\bullet i})\right)$ with tempered $\mathtt{softmax} \; \sigma_\tau(\cdot)$ (temperature $\tau$) normalising over the unspecified ($\bullet$) dimension.

## 4 MODEL

A particularly interesting characteristic of learning with explicitly structured models such as Self-StrAE or even earlier models such as the IORNN (Le & Zuidema, 2014) is the dichotomy between the upward and downward embeddings. Given their construction, the upward embeddings are always *locally-contextual*: they only encapsulate the context of the span they cover. For example, following Fig. 1, the upward embedding $\bar{e}$ for the span ‹ate doughnuts› is always the same regardless of context, no matter who did the eating. In contrast, downward embeddings are always *globally-contextual*: they must encapsulate the surrounding context by virtue of being decomposed from larger spans. For our example, this implies that there are multiple downward embeddings $\underline{e}^y$ for the given span, one for each $y \in \{$Lisa, Homer, …$\}$. To learn effective embeddings then, one must *marginalise* over these different downward embeddings to ensure that their meaning resolves over all these contexts.

### 4.1 FROM TREES TO ENTANGLED TREES

We want to have the composition embeddings amortise over all possible contexts, and simultaneously we want all decompositions embeddings to resolve to the same thing. The representation of an entity Lisa should encapsulate everything she could possibly eat. Simultaneously if we take the average of everything she could eat we should get back to Lisa. Self-StrAE does not explicitly model this behaviour in its structure. Decomposition embeddings of the same entity only interact when we calculate the loss. On top of this, because the loss is taken over the batch, they are actually treated as false negatives to each other. Even though they are terms that ought not be pushed away, the objective ask them to be.

Our innovation here is to address both these issues together by formulating the process in terms of entangled

---

**Algorithm 1** Banyan: Entangled Compose

**Input:** Global frontier $\langle (s_n, e_n) \rangle_{n=1}^N$, compose ($\circ$), concat ($\diamond$), similarity $\text{cSIM}(e, e')$
1: $\mathcal{A} \leftarrow \langle (s_n, e_n) \rangle_{n=1}^N$     ▷ initialise frontier
2: $(\mathcal{V}, \mathcal{E}) \leftarrow (\varnothing, \varnothing)$     ▷ initialise graph
3: **while** $\exists i : s_i \diamond s_{i+1} \notin_s \mathcal{V}$ **do**
4:    $i^\star \leftarrow \arg\max_i \text{cSIM}(e_i, e_{i+1})$
      ▷ location of closest adjacent pair
5:    $e_p = \circ(e_{i^\star}, e_{i^\star+1})$    ▷ compute composition
6:    $\mathcal{V} \leftarrow \mathcal{V} \cup \{(s_{i^\star} \diamond s_{i^\star+1}, e_p)\}$
7:    $\mathcal{E} \leftarrow \mathcal{E} \cup \{p \sim i^\star, p \sim (i^\star + 1)\}$
8:    $\mathcal{J} \leftarrow \{j : (s_j, s_{j+1}) = (s_{i^\star}, s_{i^\star+1})\}$
      ▷ locations of all occurrences of this pair
9:    $\mathcal{A} \leftarrow \mathcal{A} \setminus \{\forall_{j \in \mathcal{J}} \, \mathcal{A}_j, \mathcal{A}_{j+1}\}$
      ▷ delete occurrences from those locations
10:    $\mathcal{A} \leftarrow \mathcal{A} \cup_{\mathcal{J}} \{(s_{i^\star} \diamond s_{i^\star+1}, e_p)\}$
      ▷ insert composition into those locations
11: **return** Graph $(\mathcal{V}, \mathcal{E})$

---

trees—where entangling refers to reduction of a set of disjoint tree structures into a single conjoined graph structure. An example is shown in Fig. 2 with disjoint trees on the left and resulting entangled tree on the right. Here, all instances of ‹night› and ‹some are born to› are captured by a single node

representing that constituent. We call our model `Banyan` on account of this entangling, because, like the tree, it can have many roots -consisting of nodes frequently reused across contexts.

**Entangling:** Constructing an entangled tree given a set of disjoint trees is a relatively straightforward process and is formally specified in Algorithm 1. In contrast to the agglomerative clustering employed in `Self-StrAE`, here we employ a global frontier spanning all leaf nodes across the given data. The key differences to the prior methods are mainly to do with constructing a graph jointly with progressing the frontier and ensuring that new nodes are never duplicated, for which we employ a node identity $s_n$ in addition to the node embedding $e_n$.

**Incorporating context** Following the entangling of trees described, the model proceeds in a similar vein to `Self-StrAE`, by composing upwards from leaves to roots (multiple roots corresponding to multiple trees), and then decomposing downwards back to the leaves. With entangled trees, while traversing upwards each node is always composed from the same two children, but on the way back down, things are different as each separate context for a given node provides a different downward embedding. This is shown in Fig. 3 focussing on a subgraph of the entangled tree from Fig. 2(right). Note that the node in question (in blue) corresponds to the span ‹some are born to›, and has downward embeddings that incorporate context both from ‹endless night› and ‹sweet delight›. This is exactly as desired, as `Banyan` allows explicit aggregation to derive the downward embedding that resolves over the contexts. For any upward embedding $\bar{e}$ whose span occurs in different contexts $y \in \mathcal{Y}$, the corresponding downward embedding is derived by simply averaging over the different contextual down embeddings; i.e., $\underline{e} = 1/|\mathcal{Y}| \sum_y \underline{e}^y$.

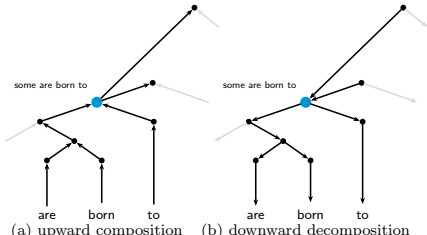

Figure 3: Upward and downward traversals for a section of the entangled tree from Fig. 2.

(a) upward composition  (b) downward decomposition

**Effectiveness and efficiency**  Beyond the ability to explicitly incorporate context across data, entangled trees also help the contrastive objective by avoiding false negatives since they do not admit duplicate nodes by construction. Furthermore, the lack of duplicate nodes also drastically impacts the memory footprint of the model as one deals with the *set* of all nodes rather than counting each instance as its own node. These effects becomes more pronounced when entangling a larger set of instances as the likelihood of false negative and duplicates goes up together.

**Practical estimation**  Given the advantages conferred by entangled trees, one would ideally want to construct it over *all* the available data. This however is not practically feasible as the size of data typically grows exponentially with time. To address this, we construct our model to estimate the given objective by taking steps over *batches* of data that are of a more manageable size, noting that this estimator is unbiased. To see this is the case, note that entangled trees only affects the downward embeddings directly, and that batching simply means that the resolved embedding is an average over *samples* instead of over all the data (*population*)—the sample mean is always an unbiased estimator of the population mean.

## 4.2 SIMPLIFIED MESSAGE PASSING

Complementary to the development of entangled trees to incorporate context, we also explore avenues to improve the message passing with the composition ($C$) and decomposition ($D$) functions. The original formulations of these from Eqs. (2) and (3) employ concatenation and splitting along with simple single-layer linear neural networks. The authors found that these simpler formulations led to better representations than e.g., Tree-LSTM cells, because they forced the model to conform to the compression order of the structure.

But if all we need for success is to respect the compression order, then we could possibly do better with an even simpler solution that exploits diagonalised functions (Ba et al., 2016)? These have become a hallmark of the recent resurgence in recurrent neural networks (Peng et al., 2023; Orvieto et al., 2023; De et al., 2024), by introducing decayed memory in the temporal dimension. Such a parameterisation means that rather than using full matrices as our $C$ and $D$ functions, we instead define them as:

$$C(\bar{e}_i, \bar{e}_{i+1}) = (\bar{e}_i \cdot \sigma(\Phi_l) + \bar{e}_{i+1} \cdot \sigma(\Phi_r)) + \phi \quad \Phi_l, \Phi_r, \phi \in \mathbb{R}^U \tag{5}$$

$$D(\underline{e}_i) = \left(\underline{e}_i \cdot \sigma(\Theta_l) + \theta_l, \quad \underline{e}_i \cdot \sigma(\Theta_r) + \theta_r\right) \quad \Theta_l, \Theta_r, \theta_l, \theta_r \in \mathbb{R}^U \tag{6}$$

with sigmoid non-linearity ($\sigma$) applied to parameters both for numerical stability and to make the functions enforce a decayed memory over structure depth. The repeated application of the diagonal composition function will decay the influence of nodes further down in the tree, thereby respecting the compression order of the structure. In addition, during composition parent representations can increase in magnitude as they are the sum of the two children. During decomposition child representations will, by necessity, reduce back down in magnitude towards the core. In this way the functions further mimic the information flow specified by the entangled trees.

These relatively simple changes have a pretty drastic effect, both in terms of performance (see experiments) as well as memory footprint, with parameters now reduced by a factor of $U$ compared to the functions from Eqs. (2) and (3)

## 5 Experiments:

### 5.1 Warmup: English language evaluation

**Goal:** Having outlined Banyan, we want to test whether it can efficiently learn semantics. We start by evaluating on English, as there are far more test sets available than for low resource languages.

**Evaluation:** We want to evaluate how well Banyan is able to learn effective semantic representations. Ideally we want to probe this at different levels of hierarchy, covering both the lexical and sentential level. Our evaluation is unsupervised, both to directly probe the effect of pretraining with the inductive bias, and because this setting has greater parity to what may be expected in a low resource domain, where there are few labelled datasets. For these reasons, we turn to a series of tasks which measure correlation between cosine similarity of embedding pairs for two examples and human judgements of their semantic correspondence. On the word level, we use Simlex-999 (Hill et al., 2015) and WordSim-S/R (Agirre et al., 2009). All tasks measure semantics, but do so on differing axes. To understand this, we must first qualify the difference between semantic similarity and relatedness. Semantic similarity measures the extent to which entities act the same way. For example, 'running' and 'singing' are similar as they share the role verb. Semantic relatedness measures conceptual association. For example, 'singing' and 'fame' may be highly related. Simlex measures similarity at the exclusion of relatedness. Wordsim S measures similarity without penalising relatedness. And Wordsim R measures relatedness. On the sentence level, we use STS-12 through 16 (Agirre et al., 2012; 2013; 2014; 2015; 2016), the STS-B (Cer et al., 2017), SICK-R (Marelli et al., 2014) and SemRel (Ousidhoum et al., 2024). Each measures slightly different aspects of sentential semantics, covering similarity, relatedness, equality and entailment. A good model should do well on all of them.

**Baselines:** We compare against the `Self-StrAE`, `GloVe` embeddings (Pennington et al., 2014) and a `RoBERTa` (Liu et al., 2019) in the medium configuration from (Turc et al., 2019). `Self-StrAE` stands as the closest point of comparison to Banyan. `Self-StrAE` indicates the performance level of structured representation learning lies, as well as any improvements we are able to achieve. `GloVe` lets us compare to traditional static embeddings. This comparison probes whether our model is learning anything more than just simple bag of word features. To obtain sentence embeddings, we report results using both the simple average of the word embeddings and the average with filler words removed following (Reimers & Gurevych, 2019). These filler words contribute little semantic information and their removal has been shown to improve performance. For `RoBERTa`, we report results using both the standard model, and again after enhancing `RoBERTa` through an extra round of contrastive `SimCSE` training (Gao et al., 2021), as a further STS baseline. In both cases, we generate sentence embeddings through mean pooling. To produce static embeddings from `RoBERTa` to use in lexical evaluation, we follow Bommasani et al. (2020) and average the contextualised representations of all occurrences of the word in the training set. The `RoBERTa` is intended as a stronger baseline. It has significantly more parameters than Banyan and is able to model meaning in context unlike `GloVe`.

**Hyperparameters and Pre-training Details:** For all models we set the embedding size to 256. For `Self-StrAE` we use the configuration of (Opper et al., 2023) and set embeddings as square matrices (i.e., $K$=16 and $U$=16). For Banyan we set these values to $K$=128 and $U$=2, because the more

Table 1: Sentence level results for models pretrained on English. Higher is better. Results represent the average across four random initialisations. Only columns where there is no standard deviation overlap between models are bolded. Spearman's $\rho$ is * 100 following convention.

| Model | STS-12 | STS-13 | STS-14 | STS-15 | STS-16 | STS-B | SICK-R | SemRel | Score |
|---|---|---|---|---|---|---|---|---|---|
| Self-StrAE | $31.98 \pm 0.58$ | $53.88 \pm 0.68$ | $37.73 \pm 0.70$ | $55.23 \pm 0.58$ | $55.55 \pm 0.47$ | $39.53 \pm 1.61$ | $51.78 \pm 0.29$ | $50.05 \pm 0.92$ | $46.59 \pm 0.43$ |
| GloVe + stopword rm | $31.61 \pm 0.31$ $39.00 \pm 0.57$ | $21.69 \pm 0.12$ $41.61 \pm 0.19$ | $27.37 \pm 0.10$ $39.31 \pm 0.18$ | $40.42 \pm 0.09$ $51.06 \pm 0.35$ | $29.27 \pm 0.12$ $45.14 \pm 0.14$ | $28.25 \pm 0.08$ $48.40 \pm 0.07$ | $50.20 \pm 0.25$ $52.80 \pm 0.04$ | $41.20 \pm 0.43$ $42.37 \pm 0.13$ | $33.75 \pm 0.04$ $44.96 \pm 0.10$ |
| RoBERTa + SimCSE | $42.77 \pm 1.27$ $50.63 \pm 1.45$ | $51.70 \pm 1.30$ $62.23 \pm 2.51$ | $45.67 \pm 1.42$ $54.17 \pm 2.10$ | $63.97 \pm 0.81$ $68.77 \pm 3.00$ | $59.60 \pm 0.61$ $66.67 \pm 1.40$ | $39.97 \pm 0.95$ $53.53 \pm 1.18$ | $52.93 \pm 0.23$ $\mathbf{56.87 \pm 1.16}$ | $52.73 \pm 0.58$ $59.27 \pm 0.93$ | $51.08 \pm 0.61$ $59.02 \pm 1.45$ |
| Banyan | $51.20 \pm 0.007$ | $\mathbf{69.10 \pm 0.002}$ | $\mathbf{63.20 \pm 0.004}$ | $\mathbf{73.20 \pm 0.002}$ | $66.60 \pm 0.002$ | $\mathbf{61.50 \pm 0.002}$ | $55.50 \pm 0.003$ | $\mathbf{61.60 \pm 0.002}$ | $\mathbf{62.70 \pm 0.001}$ |

independent channels we allowed the better the model seemed to perform. We refer the reader to the Appendix for ablations. We also note that because we can perform this reduction in channel size, the number of non-embedding parameters for Banyan drops to just 14, as these are directly proportional to $U$. We trained Self-StrAE and Banyan for 15 epochs (circa 15k steps and sufficient for convergence) using the Adam optimizer (Kingma & Ba, 2015), with a learning rate of 1e-3 for Banyan and 1e-4 for Self-StrAE using a batch size of 512. We applied dropout of 0.2 on the embeddings and 0.1 on the composition and decomposition function outputs. The temperature hyper-parameter for the Self-StrAE was set to 0.2. To process the graphs we used DGL (Wang et al., 2020). The GloVe baseline was trained for 15 epochs with a learning rate of 1e-3, and a window size of 10. We used the official C++ implementation. RoBERTa medium was trained for 200,000 steps, (10% of which were used for warmup). We used a learning rate of 5e-5, and a linear schedule. Positional embeddings are relative key-query. The configuration for RoBERTa medium is 8 layers, 8 attention heads and 2048 dimensional feedforward layers. We used the Transformers library to implement and train the model (Wolf et al., 2020). For SimCSE training, we used the default parameters and the official implementation for unsupervised RoBERTa training from Gao et al. (2021). As our pre-training corpus we selected the WikiText-103 benchmark dataset (Merity et al., 2016). The RoBERTa and GloVe baselines are trained on the full corpus (103 million tokens), representing the upper-middle end of the level of data scale that might be available for a language. Whereas we trained Self-StrAE and Banyan on a uniform subsample of 10 million tokens, representing the lower end of how many tokens might be available, because these explicit structure models are supposed to be efficient learners.

**Results:** Results are shown in Tables 1 and 2. On both the word level and sentence level Banyan does much better than Self-StrAE. We ablate the reasons for this in more detail later in the manuscript. Both models suffer on SimLex because they need to model both similarity and relatedness as the latter dictates merge (related

Table 2: Word level results analogous to Table 1.

| Model | Simlex | Wordsim-S | Wordsim-R | Score |
|---|---|---|---|---|
| Self-StrAE | $13.80 \pm 0.41$ | $54.38 \pm 0.78$ | $52.85 \pm 1.27$ | $40.34 \pm 0.66$ |
| GloVe | $27.47 \pm 0.25$ | $62.53 \pm 0.42$ | $51.00 \pm 0.56$ | $47.00 \pm 0.38$ |
| RoBERTa | $\mathbf{29.23 \pm 0.64}$ | $61.97 \pm 2.38$ | $46.00 \pm 2.13$ | $45.73 \pm 1.71$ |
| Banyan | $16.57 \pm 0.02$ | $\mathbf{63.25 \pm 0.03}$ | $\mathbf{69.00 \pm 0.01}$ | $\mathbf{49.61 \pm 0.02}$ |

concepts often compose together). However, the important thing to note is that the structured models effectively transfer the same performance from the word level to the sentence level. They can take advantage of composition, and transfer the meaning of the parts to understanding the meaning of the whole. The GloVe baseline is good on the word level, but does not generalise to the sentence level as well as the transformer, even when we give it stopword removal. It cannot transfer semantic knowledge seamlessly to different levels of complexity. Banyan can, and is able to achieve comparable or better performance than the SimCSE RoBERTa despite being much smaller and exposed to 10x less pre-training data. This means we have a structured model that remains efficient and cheap, and also effective at representation learning.

## 5.2 MULTILINGUAL EVALUATION:

**Goal:** From the results in English we know that Banyan is an efficient learner: it can produce good representations without requiring large-scale data or compute. This implies potential use for under represented communities, whose languages are not well covered by current NLP approaches. Now we have to test that.

**Evaluation:** Learning semantic representations for low resource languages remains an ongoing challenge in NLP. A core problem is not just the lack of training data, but also the lack of evaluation datasets. Recent work by Ousidhoum et al. (2024) has sought to address this issue, providing semantic relatedness test sets for several low resource Asian and African languages. These test sets are evaluated the same as before, comparing the cosine similarity between model embeddings for

Table 3: Multilingual Results. Banyan performance is taken over four random seeds. Baselines marked with † have been finetuned on supervised semantic similarity datasets. FT denotes unsupervised finetuning using masked language modelling on the same corpora as Banyan.

| Model | Indonesian | Arabic | Telugu | Marathi | Mor. Arabic | Kinyarwanda | Hausa | Afrikaans | Spanish | Amharic | Hindi | Score |
|---|---|---|---|---|---|---|---|---|---|---|---|---|
| XLM-R | 46.7 | 31.6 | 46.3 | 55.7 | 17.4 | 13.2 | 4.1 | 56.2 | 68.9 | 57.3 | 52.7 | 40.92 |
| Llama-3.1 (8B) | **53.4** | 31.1 | 65.6 | 63.4 | 19.4 | 19.7 | 6.1 | 65.4 | 66.7 | 64.1 | 61.7 | 46.96 |
| Mistral Nemo | 50.7 | 20.1 | 57 | 52.3 | 15.1 | 16.3 | 1.8 | 58.3 | 66.2 | 53.2 | 55.8 | 40.62 |
| MiniLM-L12† | 39 | 16.1 | 34.8 | 39.5 | 13.5 | 35 | 32.7 | 74.1 | 58.8 | 9.6 | 43.8 | 36.08 |
| Paraphrase XLM-R† | 46.1 | **61** | 58.1 | **79.6** | 7.1 | 43.2 | 22.5 | 76.8 | 71.7 | 64.6 | 52 | 52.97 |
| XLM-R (FT) | 47.9 | 33.6 | 68.8 | 75.1 | 21.6 | 19.4 | 14.6 | 72.6 | **72.8** | 59.6 | 57.6 | 49.41 |
| Banyan | $44.17 \pm 1.11$ | $43.20 \pm 1.82$ | **$71.13 \pm 0.91$** | $67.67 \pm 0.64$ | **$52.00 \pm 2.25$** | **$46.1 \pm 0.32$** | **$43.7 \pm 1.21$** | **$78.68 \pm 0.30$** | $60.95 \pm 0.76$ | **$66.18 \pm 0.46$** | $61.83 \pm 0.6$ | **57.78** |

two sequences with human judgements of their semantic match. As before, evaluation is zero-shot unsupervised. Allowing us to evaluate Banyan on Indonesian, Arabic, Telugu, Marathi, Moroccan Arabic, Kinyarwanda, Hausa, Afrikaans, Spanish, Amharic and Hindi. These represent a spectrum in terms how well resourced they are. For example, Spanish and Hindi are reasonably well represented, while Moroccan Arabic and Kinyarwanda have extremely little training data.

**Baselines:** We select XLM-R (Conneau et al., 2019): a transformer encoder trained on 2TB of multilingual data. Llama 3.1 8B (Dubey et al., 2024): a decoder only LLM trained on 15 trillion tokens. Mistral Nemo 12B: a decoder only LLM designed with multi-lingual capacities in mind. In addition we also compare against two specialised embedding models from the sentence transformers range (Reimers & Gurevych, 2019): Mini-LM-L12-V2 and Paraphrase-XLM-R-Multilingual-V1. These are pre-trained transformer encoders that have been finetuned on supervised datasets designed to produce high quality semantic representations. The baselines we select here are emblematic of the kind of models one might reach for in order to embed a corpus. For all models we use mean pooling to produce the sentence representation following Reimers & Gurevych (2019); Li & Li (2024). Finally, for parity we include an XLM-R baseline which is finetuned on the same corpora.

**Banyan Pre-training and Hyperparameters:** For Afrikaans, Spanish and Amharic we obtained corpora from Leipzig Corpora Collection[1] (Goldhahn et al., 2012). For Amharic we utilised a MiT licenced pre-training set of 1 million sequences available on the Huggingface hub at this link. Kinyarwanda and Hausa data was sourced from Opus (Nygaard & Tiedemann, 2003). Each dataset consists of roughly 10 million tokens. We utilise a pre-trained BPE tokenizer for each language from the BPEMB Python package (Heinzerling & Strube, 2018). Though the package also provides pre-trained embeddings, we solely use the tokenizer and learn embeddings from scratch. For the model hyperparameters we keep all the settings from the experiments on English. For XLM-R we finetune for up to 100k steps with early stopping, using a linearly scheduled learning rate of 5e-5 with 10 percent of stepping serving as warmup. XLM-R runs at batch size 128 across 4xA40 45gb cards.

**Results:** See Table 3. In Spanish, a well resourced language with high coverage, the transformer baselines almost all outperform Banyan. However, as languages become lower resourced the picture changes, and Banyan outperforms or is comparable to the baselines. This even includes the multilingual XLM-R that has undergone supervised training to produce better representations. While finetuning XLM-R improves performance the amount of benefit it provides is not uniform and is insufficient to prove viable in the very low resource cases. Banyan is able to learn competitive representations consistently across languages, unsupervised and with very little data, meaning it provides a viable alternative for producing embeddings cheaply and efficiently for low resource languages.

## 5.3 EFFICIENCY

Alongside its embedding matrix, Banyan has two central components: the composition and decomposition functions. We diagonalise these functions so that they are both easier to compute and have fewer parameters than standard weight matrices, $(2U$ rather than $2U \times U)$, achieving a further order of magnitude reduction in parameters compared with the already minimal Self-StrAE.

Secondly, by exploiting entangled tree structure the number of nodes grows at a significantly reduced rate with batch size compared with standard sentential trees (see

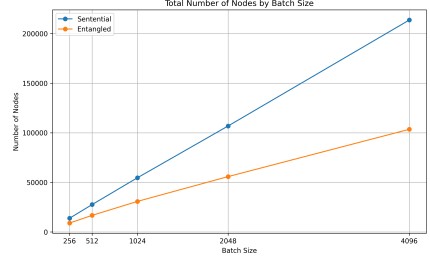

Figure 4: Total #nodes in entangled trees vs sentential trees as batch size grows.

---

[1]For Spanish and Hindi we select the mixed corpus and uniformly subsample to reduce size to ≈10M tokens.

Table 4: Ablations of modelling changes made for Banyan. Higher is better. Results represent the average across four random initialisations. Only columns where there is no standard deviation overlap between models are bolded. Spearman's $\rho$ is * 100 following convention.

| Model | STS-12 | STS-13 | STS-14 | STS-15 | STS-16 | STS-B | SICK-R | SemRel | Score |
|---|---|---|---|---|---|---|---|---|---|
| Standard Trees | $31.98 \pm 0.58$ | $53.88 \pm 0.68$ | $37.73 \pm 0.70$ | $55.23 \pm 0.58$ | $55.55 \pm 0.47$ | $39.53 \pm 1.61$ | $51.78 \pm 0.29$ | $50.05 \pm 0.92$ | $46.59 \pm 0.43$ |
| + diag functions | $35.13 \pm 0.33$ | $56.05 \pm 0.24$ | $40.58 \pm 0.05$ | $58.83 \pm 0.10$ | $56.78 \pm 0.21$ | $44.10 \pm 0.14$ | $53.35 \pm 0.17$ | $52.65 \pm 0.17$ | $49.68 \pm 0.06$ |
| ++ CE loss | $47.10 \pm 1.04$ | $61.85 \pm 1.44$ | $58.60 \pm 1.34$ | $70.45 \pm 0.57$ | $62.45 \pm 0.70$ | $59.50 \pm 0.53$ | $\mathbf{59.00 \pm 0.26}$ | $60.33 \pm 0.26$ | $59.91 \pm 0.54$ |
| Entangled Trees | $38.98 \pm 0.39$ | $61.75 \pm 0.14$ | $43.65 \pm 0.46$ | $58.21 \pm 0.41$ | $55.29 \pm 0.23$ | $46.15 \pm 0.71$ | $53.93 \pm 0.16$ | $52.53 \pm 0.09$ | $51.31 \pm 0.13$ |
| + diag functions | $44.15 \pm 0.002$ | $62.80 \pm 0.002$ | $48.30 \pm 0.001$ | $64.60 \pm 0.002$ | $60.30 \pm 0.001$ | $49.80 \pm 0.002$ | $55.14 \pm 0.001$ | $57.70 \pm 0.001$ | $55.23 \pm 0.001$ |
| ++ CE loss | $\mathbf{51.20 \pm 0.007}$ | $\mathbf{69.10 \pm 0.002}$ | $\mathbf{63.20 \pm 0.004}$ | $\mathbf{73.20 \pm 0.002}$ | $\mathbf{66.60 \pm 0.002}$ | $\mathbf{61.50 \pm 0.002}$ | $55.50 \pm 0.003$ | $\mathbf{61.60 \pm 0.002}$ | $\mathbf{62.70 \pm 0.001}$ |

Fig. 4). This is because the number of reused constituent nodes also grows as batch size increases, and entangled trees capture the set of all constituents, which consequently does not grow as drastically. In practical terms, because entangled trees requires fewer nodes, and each node requires two distinct embeddings ($\bar{e}$ and $\underline{e}$) to be held for it, reducing the number of nodes required leads to radical improvements in memory efficiency. Put together, these changes mean that we can train Banyan very quickly as we can use large batches and its small number of parameters ensure quick convergence. On a single Nvidia A40 GPU with a batch size of 1024, Banyan pretrains from scratch in under 50 minutes, meaning that the total cost of pretraining a Banyan model sits at around 30 cents[2]. Free-tier Google Colab users can achieve similar results in about two hours with a smaller batch size. Inference can also be performed on CPU on typical laptops, because the model is so small. Combined with its data efficiency, we believe this provides a promising alternative for low resource languages and communities.

## 5.4 ABLATIONS

We have shown that Banyan is more effective than its `Self-StrAE` predecessor, but what is the impact of the different modelling changes we made? To test this we ablate our results from our first set of experiments on English (see Table 4).

The simplest positive impacts to see are from the introduction of the diagonalised composition and decomposition functions. These are sigmoided scalar values with which we multiply embeddings. Therefore they act similarly to the fast weights of Ba et al. (2016), decaying in the influence of embeddings further down in the tree on the root representation. This means that the embeddings produced by the model are restricted to conform to the compression order dictated by the structure, and we know from Opper et al. (2023), that the more we can enforce this constraint the better our representations will end up. Secondly, such simple message passing functions bias the representation space towards informative separability. There has to be some signal from which to perform reconstruction, and all the pressure is now on the representations.

Switching the objective to cross entropy over the vocabulary rather than the contrastive objective used by Opper et al. (2023) also yields significant benefits. This is likely because the contrastive loss is supposed to be beneficial because it enforces a pressure for representations to be uniformly distributed in space (Wang & Isola, 2020). However, our other modelling changes already push towards this quality. While it is a shame because the contrastive loss is conceptually elegant. It is known to have problems and eventually lead to shortcut solutions (Robinson et al., 2021). Therefore having a more robust objective grounded in data, like the cross entropy over the vocabulary, is actually quite nice.

Finally, changing to entangled trees is also beneficial. The effect is more pronounced before switching to the cross entropy objective, as it removes the issue of false negatives as discussed in Section 4. However, it also is beneficial beyond this. Entangling explicitly allows the model to take advantage of shared constituency structure between complex sequences, because it combines the information from all incoming parent messages. The fact that performance improves using the cross entropy objective shows that explicitly allowing the model to take advantage of such systematicity is useful.

Table 5: Number of non-embedding parameters for models studied.

| Model | Banyan | Self-StrAE | RoBERTa (M) | All-MiniLM-L12-V2 | XLM-R | Llama 3.1 | Mistral Nemo |
|---|---|---|---|---|---|---|---|
| Params | 14 | 1072 | $\approx$10M | $\approx$21M | $\approx$85M | $\approx$8B | $\approx$12B |

---

[2]Current cloud computing costs sourced from: https://www.runpod.io/pricing

## 6 CONCLUSION, LIMITATIONS AND FUTURE WORK

We introduce Banyan, a Self-Structuring AutoEncoder. Banyan's focus on global, entangled structure and simplified message passing exploits the benefits of structured compositions inherent in language data. It is more effective and efficient than prior work from which we draw three central conclusions.

Firstly, explicitly modelling structured compositions is an effective inductive bias. Table 5 shows the parameters for the structured models versus the baselines. The structured models are far smaller, with tens or thousands of parameters instead of millions or billions. And nonetheless, Banyan is still competitive across several metrics, indicating we have found an efficient learning procedure.

Secondly, we have not yet fully exploited the potential of the inductive bias. Banyan still relies on greedy agglomerative clustering to induce structure. This is effective, but sub-optimal. Future work could make the structure induction procedure parametric and learnable. The type of structure models are exposed to impacts the quality of learnt semantic representations (Opper et al., 2023). So if *how* we induce structure improves, the model should learn significantly better representations.

Thirdly, while this paper focuses on recursive neural networks for the purposes of efficiency and low resource applicability, the method could in principle be applied to representations from pre-trained transformer models. Firstly, the transformers attention essentially defines a soft (fully connected) graph between tokens, which could serve as a more flexible basis for constructing Banyan's discrete structures. Moreover, the entangled tree structure essentially serves as a map of the conceptual associations learned by a model, and could provide an interesting probe into the representation space of pre-trained LLMs.

Finally, good and cheap embedding models are useful for many applications. For example, the digital humanities need to organise corpora of ancient languages, making it easier for researchers to access texts they need. But these corpora are small, and these languages are unlikely to be present in pretraining corpora of larger models. Banyan provides an efficient solution for producing representations for both these use cases and low resource languages and under represented communities more generally. To conclude, Banyan addresses the problem of efficient learning in low-resource settings.

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

# A APPENDIX

## A.1 THE $k$ AND $u$ BALANCE

The change to diagonal composition functions allows us to reduce the number of total parameters while maintaining performance. This is because the number of parameters is directly proportional to channel size $u$. We show ablations for this finding in Table 6. Our findings are similar to those of Opper & Narayanaswamy (2024) the smaller the channel size the better the model performs, although in our case we keep things stable between seeds whereas for them when they simplified they faced issues with extreme instability during training. This is thanks to the new message passing functions.

Table 6: Performance Depending on $k$ and $u$ values using new functions. Scores are the average of four random seeds.

| $k$ | $u$ | Lex Score | STS Score |
|---|---|---|---|
| 4 | 64 | $42.9 \pm 0.01$ | $43.5 \pm 0.04$ |
| 8 | 32 | $43.2 \pm 0.02$ | $48.6 \pm 0.01$ |
| 16 | 16 | $47.02 \pm 0.03$ | $62.2 \pm 0.01$ |
| 32 | 8 | $49.2 \pm 0.01$ | $62.9 \pm 0.01$ |
| 64 | 4 | $48.7 \pm 0.01$ | $62.9 \pm 0.01$ |
| 128 | 2 | $49.61 \pm 0.02$ | $62.7 \pm 0.001$ |
| 256 | 1 | $48.7 \pm 0.01$ | $62.9 \pm 0.001$ |

