# OpenReview forum: "Banyan: Improved Representation Learning with Explicit Structure"
_ICLR.cc/2025/Conference — Submitted to ICLR 2025_

### Official Review · Reviewer_ryjZ · 2024-10-22

**Soundness:** 3
**Presentation:** 2
**Contribution:** 3
**Rating:** 8
**Confidence:** 2

**Summary:**

This paper introduces a new recursive graph neural network for learning text representations in low-resource languages: Banyan (like the tree). This model extends previous work by building nested trees over sequences that share the same tokens. In Banyan the same tokens will have the same tree node, even if it comes from different sequences. For scalability reasons, the trees are constructed from a batch of samples rather than from an entire dataset. Embeddings are learned from a simplified message passing algorithm that traverse the trees both in bottom-up and top-down directions.
Having nested trees provides multiple advantages, notably the reduction of duplicated nodes and multiple context representations within the same node (in downward embeddings).
These advantages translate to strong semantic representations in both English (when compared to RoBERTa) and lower-resourced languages (when compared to XLM-R, Llama 3.1, Mistral).

**Strengths:**

This paper introduces a novel recursive model that learns textual representations and its learning mechanism.
The proposed architecture is novel and seems promising as it yields good results when compared to other more classical methods.
In addition, the proposed method is very efficient: it requires very little training and has only 14 non-embedding parameters.

**Weaknesses:**

The task being tackled is not clear from the abstract or the introduction. The motivation is well described (learning powerful text representations for under-resourced languages), but the task used to evaluate the Banyan model (Semantic Textual Similarity - STS task) is not described. The paper could gain clarity by mentioning the task earlier.

Evaluation is based on cosine similarity between sentences compared to human judgment. The paper would attract greater significance if the representations from the proposed model were tested in targeted applications such as sentiment classification or machine translation.

**Questions:**

Previous approaches used 1 tree per sequence. Banyan uses one tree for all sequences sharing the same tokens in a batch.
Some analysis of scale would be nice to complement Figure 4 in the paper. For instance, how many trees do you usually have in an entire batch? How many (sub-)sequences are represented in 1 tree?

---

> ### Author Response · Authors · 2024-11-18
>
> Thank you for your review and positive feedback on our paper! We have edited the manuscript to introduce the task earlier and will be adding an appendix containing the scale analysis you requested. A brief overview of the analysis so far is:
>
> We usually run at batch size 512 (though larger is possible). Each tree covers a sequence of an average length of 30 and therefore has 29 non-terminal nodes. Non-entangled this means that the collection of tree would have about 33k nodes, while the entangled tree consists of about 18k so we have roughly a 50% duplicate rate. Interestingly the amount of reuse for a node follows a roughly Zipfian distribution which means that this law applies also to consituent structures.
>
> We appreciate your constructive feedback, if there are any further questions you would like us to answer please let us know!

---

> > ### Comment · Reviewer_ryjZ · 2024-11-20
> > **acknowledgement**
> >
> > thanks for addressing my comments. No further questions at this time.

---

### Official Review · Reviewer_gpX7 · 2024-10-26

**Soundness:** 3
**Presentation:** 2
**Contribution:** 2
**Rating:** 6
**Confidence:** 4

**Summary:**

The paper presents Banyan, an efficient and lightweight framework for representation learning on low resource languages. The method has been evaluated in common English tests as well as a series of low-resource languages. This method demonstrates great performance as well as remarkable efficiency. While the technical innovations are impressive, there are still several remaining concerns.

**Strengths:**

This proposed method is technically solid, and makes substantial improvement compared with existing methods.
The proposed method is highly efficient and lightweight, especially when compared with Large language models.

**Weaknesses:**

Some evaluation parts still need further validation. The details are described in questions

**Questions:**

1. Regarding low resource language evaluation, the authors only used four languages, while there are many other languages in the released semeval dataset, is there any cherry-picking on the languages evaluated ? How will the model perform on other low resource languages ?
2. The authors trained Banyan on datasets from Leipzig Corpora Collection, is there any overlap between the training corpora and the testing dataset ? and if the baseline methods such as XLM-R are also pre-trained on these corpora, how will the models perform ?

---

> ### Author Response · Authors · 2024-11-18
>
> Thank you for your feedback regarding our paper! We are glad that you found our ‘technical innovations impressive’ and recognise the ‘great performance and remarkable efficiency’ of our method. We also appreciate the concerns you raised and agree that addressing them will help to significantly strengthen our work. We have updated the manuscript and provide a general response below:
>
>
> **More Languages:** We do not cherry pick! Our choices were based on representing a spectrum in terms of ‘well resourcedness’ and where it looked like the test sets had reasonably good annotation. Nonetheless we recognise the concern and have expanded our evaluation in section 5.2 to additionally include Indonesian, Arabic, Telugu, Marathi, Moroccan Arabic, Kinyarwanda and Hausa. The pattern remains largely the same as before. Generally speaking, the more low-resource the language, the more favourable the comparison becomes for Banyan. Please see results and further discussion in the updated manuscript.
>
>
> **Possible Data Leakage:** We have checked whether any test sentences appear in the pretraining corpora using lexical overlap. We did not find any exact matches or significant outliers that might indicate leakage between the two. If there are further tests you would like us to run please let us know, and we would be more than happy to do so!
>
> **Finetuning XLM-R:**  We have finetuned XLM-R on the same corpora we used to train Banyan and report results as a further baseline in section 5.2. While finetuning does improve performance (sometimes quite significantly) Banyan remains better overall, particularly in the low resource settings. We would also like to add that Banyan can be trained on a single A40 in under an hour, while a reasonable finetune of XLM-R requires 4xA40s and between 10-16 hours depending on how out-of-distribution the language is for the tokenizer. As finetuning XLM-R is such a comparatively compute intensive process  we are continuing to work on adding further random seeds to the evaluation and will be updating with standard deviations throughout the rebuttal phase. However, we ask your patience as we are on a compute constrained academic budget.
>
>
> If there are any further questions that you would like us to respond to please let us know! If you feel we have adequately addressed your concerns please consider increasing your score. Thank you for your time and constructive feedback!

---

> > ### Comment · Reviewer_gpX7 · 2024-11-22
> >
> > Thanks for your reply, looks better now.

---

### Official Review · Reviewer_Equd · 2024-11-04

**Soundness:** 3
**Presentation:** 3
**Contribution:** 3
**Rating:** 5
**Confidence:** 3

**Summary:**

In this work, the authors present a method called "Banyan" that can learn the semantic representations with hierarchical structure. It has two main improvements (entangled tree structure and diagonalized message passing functions) comparing to SELF-STRAE. According to their experimental results, Banyan achieves competitive performance with transformer based baselines. It also shows the low cost and efficiency.

**Strengths:**

1. The proposed method is a novel approach for semantic representation. Tree structure is inject into the representation.

2. According to the experiments results (Table 1 and Table 2),  Banyan achieves competitive performance with transformer based baselines on sentence level and work level. The results are also much better than previous baseline (Self-StrAE).

3. There are a clear ablations study on the effect of different modeling changes. It is helpful for others to understand the method.

**Weaknesses:**

1. For this proposed semantic representations, both the structure and representation are learned. The first  embedding tokens are used to determine which tokens to merge and in what order. The initial token embeddings have a large impact on the proposed method. It is a little unclear about this part (are they randomly initialized? Are they updated during model training? etc)

2. Only 14 non-embedding parameters are used in the proposed method. It could also limit the ability the proposed model. It would be great if the proposed method can be used in the transformer-based embedding in the future.

**Questions:**

There are composition and decomposition (corresponding up and down embeddings), what are the embedding used for sentence-level and word-level?

---

> ### Author Response · Authors · 2024-11-18
>
> Thank you for your feedback on our paper, recognising the efficiency and improvements that our method delivers! We would like to address your concerns in our response:
>
> **How are embeddings initialised:** We initialise randomly from a uniform distribution and update the embeddings during training. This means that tokens that can frequently be merged together have their representations drawn together. In turn this leads to regularities emerging in the structure which the model can exploit for better reconstruction. Despite the fact that the model is randomly initialised this approach leads to consistent patterns forming and performance is stable across initialisations. We have attempted to highlight this in lines 136-141, please let us know if anything remains unclear!
>
>
> **Parameter Scale:** The primary goal of our research is to test whether inductive biases can help make learning more resource efficient. The low number of parameters enhances the effect of the biases and helps drive efficiency. We agree that the question of whether the method can be applied to transformers is interesting, but we think it falls outside of the current scope of our work. Particularly because one of the advantages of Banyan is that it can be run very cheaply. However, we have added a section to the conclusion where we discuss potential avenues for incorporating the technique with transformers (lines 501-506), because we definitely agree it is an interesting question for future work. Please let us know if you think this is an acceptable compromise and if there are any further questions or suggestions you have for the section.
>
>
> **Which embeddings are used:** Currently we only use the up embedding as this represents the semantics of the span. However there are a lot of potential applications involving the down embeddings such as NER, IR etc. which could be investigated in future.
>
> If you have any further questions or concerns please let us know. We are committed to engaging with you throughout the rebuttal process and appreciate your feedback! If you feel we have adequately addressed your concerns please consider raising your score.

---

> > ### Author Response · Authors · 2024-11-25
> >
> > Dear Reviewer Equd,
> >
> > We hope our response has addressed your concerns, please let us know if you have any further questions or recommendations for us to respond to. Given that we are reaching the end of discussion period we want to make sure we have time to incorporate your feedback.
> >
> > Best,
> >
> > The Authors

---

### Official Review · Reviewer_CFgK · 2024-11-04

**Soundness:** 2
**Presentation:** 2
**Contribution:** 2
**Rating:** 3
**Confidence:** 4

**Summary:**

This paper presents a strategy of representation learning by utilizing structural information discovered during learning. The proposed work is built upon a prior work with two specific pieces of improvement on building the structures and propagating the information during representation learning. Empirical evaluation was performed on multiple NLP tasks.

**Strengths:**

- The proposed new strategies for constructing tree structures are intuitive
- The proposed framework is parameter efficient and easy to learn.

**Weaknesses:**

- The writing of this paper can be significantly improved. There are some inconsistent statements and inconsistent terminology, which should be easily fixed after proofreading. For example, the paper uses both "under resourced languages" and "under represented languages", which I think it should be "low-resource languages".
- There are some unsupported claims, for example, in the first section, the claim "It is thought that this principle lets humans generation ..." should be supported with references from prior work.
- There are also some technical details that are unclear in the paper. For example, in line 204 - 207, what are criteria of entangling different trees together, and what are the benefits of entangling subtrees together?
- There is a minor issue with the citation format; it should be, for example, (Tai et al., 2015).
- There are some existing works along the line of upward and downward along a tree structure to learn representation, which are not discussed in this paper. For example, Ji and Eisenstein, One Vector is Not Enough: Entity-Augmented Distributed Semantics for Discourse Relations, 2015.
- The experiments are not sufficient; for example, some recent large language models are not included in the comparison.

**Questions:**

Please refer to the comments in the weaknesses section.

---

> ### Author Response · Authors · 2024-11-18
>
> Thank you for your feedback regarding the paper, we have updated the manuscript as follows:
>
> **Terminological Consistency and Citation Format:**  Thank you for pointing these out, we have edited them to be consistent in the updated manuscript.
>
> **Composition for efficiency and generalisation:**
> Thank you for pointing this out; we have included the following relevant references in the updated manuscript. The notion that the principle of compositionality allows humans to generalise is a long-standing one, sometimes formally referred to as systematic compositionality (Fodor & Pylyshn, 1988) [1], and involves the ***infinite use of finite means*** (Chomsky, 1965) [2].
> In more recent years, this has been the subject of a range of papers including formalisation (e.g. Wiedemer et al 2023) [3], analysis (e.g. Ito et al, 2022) [4], and modelling (e.g. Lake et al, 2017) [5].
>
> **Entangling Criteria:** The criterion for entangling trees together is when they both contain instances of the same node (e.g. ‘some are born to’ as illustrated in Figure 2). The benefits include both a significant increase in memory efficiency (see section 5.3), as well as explicitly tying together higher order nodes that have the same constituency structure (225-239 for description and 5.4 for ablation of effectiveness).
>
> **Prior Tree Models:** Thank you for pointing out this reference; we have included it in our related work.
> However, we note that their innovation appears to be in the task specific application and not the architecture itself. The model is in fact derived from the work Socher et al. [6,7] , and is identical to (and preceeded by) the IORNN [8]---all of which we cite in our work. The work done in Ji and Eisenstein, 2015 does not preclude any of the contributions in this current work. Note also, that Self-StrAE baseline we use already outperforms the IORNN and Tree-LSTM [9] and Banyan is a significant improvement over Self-StrAE.
>
> **Missing Recent LLMs:**  We included **Llama 3.1 and Mistral Nemo** which at time of writing were very new releases, but we are happy to include additional LLMs. Which large language models in particular do you have in mind?
>
>
> Do let us know if you have any additional concerns regarding the paper, we look forward to engaging with you during the discussion period!
>
>
> References:
>
> [1] Jerry A. Fodor and Zenon W. Pylyshyn. Connectionism and cognitive architecture: A critical
> analysis. Cognition, 28(1):3–71, March 1988
>
> [2] Noam Chomsky. Aspects of the Theory of Syntax, 1965
>
> [3] Compositional Generalization from First Principles
> Thaddäus Wiedemer, Prasanna Mayilvahanan, Matthias Bethge, Wieland Brendel, 2023
>
> [4] Compositional generalization through abstract representations in human and artificial neural networks
> Takuya Ito, Tim Klinger, Douglas H. Schultz, John D. Murray, Michael W. Cole, Mattia Rigotti, 2022
>
> [5] Building machines that learn and think like people
> Brenden M Lake, Tomer D Ullman, Joshua B Tenenbaum, Samuel J Gershman, 2017
>
> [6] Richard Socher, Jeffrey Pennington, Eric H. Huang, Andrew Y. Ng, and Christopher D. Manning.
> Semi-supervised recursive autoencoders for predicting sentiment distributions. In Empirical
> Methods in Natural Language Processing (EMNLP), pp. 151–161, 2011.
>
> [7] Richard Socher, Alex Perelygin, Jean Wu, Jason Chuang, Christopher D. Manning, Andrew Ng, and
> Christopher Potts. Recursive deep models for semantic compositionality over a sentiment treebank.
> In Empirical Methods in Natural Language Processing (EMNLP), pp. 1631–1642, 2013.
>
> [8] Phong Le and Willem Zuidema. Inside-outside semantics: A framework for neural models of
> semantic composition. In NIPS 2014 Workshop on Deep Learning and Representation Learning,
> 2014
>
> [9] Mattia Opper, Victor Prokhorov, and Siddharth N. Strae: Autoencoding for pre-trained embeddings
> using explicit structure. In Proceedings of the 2023 Conference on Empirical Methods in Natural
> Language Processing, pp. 7544–7560

---

> > ### Author Response · Authors · 2024-11-25
> >
> > Dear Reviewer CFgK,
> >
> > We hope our response has addressed your concerns, please let us know if you have any further questions or recommendations for us to respond to. Given that we are reaching the end of discussion period we want to make sure we have time to incorporate your feedback.
> >
> > Best,
> > The Authors

---

### Meta-Review · Area_Chair_5Mu2 · 2024-12-22

**Metareview:**

This paper proposes a recursive autoencoder for learning text representations. The method works by recursively merging adjacent embeddings from bottom up to build a tree, and then top down splitting to reconstruct leaf embeddings. Experiments on semantic text similarity demonstrate the effectiveness of the proposed approach.

Strengths:
1. The method itself is simple and provides an alternative to existing representation learning methods by focusing on structures.

Weaknesses:
1. The evaluation is mainly conducted using semantic text similarity but not directly downstream applications such as machine translation or sentiment analysis, or better yet, the GLUE benchmark which is often used for evaluating representation learning methods.

Overall, while this is a very interesting method, the evaluation itself is weak. I think this paper can be significantly improved by evaluating it on downstream applications. I'm recommending rejection for the current version, but I wouldn't mind if the paper gets accepted.

**Additional Comments On Reviewer Discussion:**

Most reviewers' questions are clarification questions that have been addressed by authors. However, reviewer ryjZ's point on limited evaluation has not been addressed yet and I think that review provides constructive feedback that can be incorporated into the next version of this work.

---

### Decision · Program_Chairs · 2025-01-22

Reject